# Optimizing Observation Plans for Identifying Faxon Fir (*Abies fargesii* var. *Faxoniana*) Using Monthly Unmanned Aerial Vehicle Imagery

Weibo Shi [1,2], Xiaohan Liao [2,3,*], Jia Sun [1], Zhengjian Zhang [4,5], Dongliang Wang [2,3], Shaoqiang Wang [1,6,7], Wenqiu Qu [2,3], Hongbo He [2,3], Huping Ye [2,3], Huanyin Yue [2,3] and Torbern Tagesson [8,9]

1   Hubei Key Laboratory of Regional Ecology and Environment Change, School of Geography and Information Engineering, Chinese University of Geosciences, Wuhan 430074, China
2   State Key Laboratory of Resources and Environmental Information System, Institute of Geographic Sciences and Natural Resources Research, Chinese Academy of Sciences, Beijing 100101, China; wangdongliang@igsnrr.ac.cn (D.W.)
3   Key Laboratory of Low Altitude Geographic Information and Air Route, Civil Aviation Administration of China, Beijing 100101, China
4   Research Center for Digital Mountain and Remote Sensing Application, Institute of Mountain Hazards and Environment, Chinese Academy of Sciences, Chengdu 610041, China
5   Wanglang Mountain Remote Sensing Observation and Research Station of Sichuan Province, Mianyang 621000, China
6   Key Laboratory of Ecosystem Network Observation and Modeling, Institute of Geographic Sciences and Natural Resources Research, Chinese Academy of Sciences, Beijing 100101, China
7   College of Resources and Environment, University of Chinese Academy of Sciences, Beijing 100049, China
8   Department of Physical Geography and Ecosystem Science, Lund University, P.O. Box 117, SE-22100 Lund, Sweden
9   Department of Geosciences and Natural Resource Management, University of Copenhagen, 1172 Copenhagen, Denmark
*   Correspondence: liaoxh@igsnrr.ac.cn

**Abstract:** Faxon fir (*Abies fargesii* var. *faxoniana*), as a dominant tree species in the subalpine coniferous forest of Southwest China, has strict requirements regarding the temperature and humidity of the growing environment. Therefore, the dynamic and continuous monitoring of Faxon fir distribution is very important to protect this highly sensitive ecological environment. Here, we combined unmanned aerial vehicle (UAV) imagery and convolutional neural networks (CNNs) to identify Faxon fir and explored the identification capabilities of multispectral (five bands) and red-green-blue (RGB) imagery under different months. For a case study area in Wanglang Nature Reserve, Southwest China, we acquired monthly RGB and multispectral images on six occasions over the growing season. We found that the accuracy of RGB imagery varied considerably (the highest intersection over union (IoU), 83.72%, was in April and the lowest, 76.81%, was in June), while the accuracy of multispectral imagery was consistently high (IoU > 81%). In April and October, the accuracy of the RGB imagery was slightly higher than that of multispectral imagery, but for the other months, multispectral imagery was more accurate (IoU was nearly 6% higher than those of the RGB imagery for June). Adding vegetation indices (VIs) improved the accuracy of the RGB models during summer, but there was still a gap to the multispectral model. Hence, our results indicate that the optimized time of the year for identifying Faxon fir using UAV imagery is during the peak of the growing season when using a multispectral imagery. During the non-growing season, RGB imagery was no worse or even slightly better than multispectral imagery for Faxon fir identification. Our study can provide guidance for optimizing observation plans regarding data collection time and UAV loads and could further help enhance the utility of UAVs in forestry and ecological research.

**Keywords:** unmanned aerial vehicles; convolutional neural networks; tree species classification; vegetation indices; Faxon fir; forest inventory

## 1. Introduction

As one of the main tree species in the subalpine coniferous forests of western China, Faxon fir (*Abies fargesii* var. *faxoniana*) plays a key role in different regeneration niches, demographic characteristics, and responses to gap disturbance regimes [1]. Faxon fir is an endemic tree species in China that coexists with *Fargesia denudate* Yi., which is one of the most important edible bamboos for the Chinese Giant Panda in the subalpine region of Sichuan Province in southwestern China [2]. However, Faxon fir has strict requirements on the temperature and humidity of the growing environment. Global climate change may affect the natural distribution of Faxon fir, jeopardizing the stability of regional forest communities and endangering the natural habitat of Giant Pandas [3]. Therefore, mapping the distribution of Faxon fir is of high importance for monitoring this highly sensitive ecological environment.

Field surveys have often been used previously to obtain the vegetation distribution, which is time-consuming and costly [4]. For forested areas in southwest China, complex terrain conditions increase the additional safety risks of this survey method. With the continuous development of unmanned aerial vehicles (UAVs) and remote sensing technologies, UAVs have become a new type of remote sensing observation platform widely used in various fields [5–7]. They can also adapt to the complex geographical environment of forests and enable the acquisition of high temporal and spatial resolution image data, which provide strong help for the investigation and monitoring of forest resources [8,9]. However, in dense forest environments, UAVs are limited by the effects of the forest canopy and still require traditional forest surveys for monitoring. In general, UAVs are widely used in forest degradation [10], the extraction of forest structural parameters [11–13], tree species classification [14,15], pest and disease monitoring [16], and biomass estimation [17,18], etc.

High-resolution UAV imagery contains abundant spatial or spectral information [5], which also increases inter- and intra-class variability in features. Traditional pixel-based classification usually solely utilizes the spectral features of UAV imagery, leading to classification results with salt-and-pepper noise [19]. On the other hand, object-based image analysis considers spatial contextual features. Additionally, these spatial features are limited to the interior of segmentation object. This means that there is little consideration of the global and contextual features of UAV imagery [20,21]. With the rise of deep learning, convolutional neural networks (CNNs) have made significant progress in computer vision and pattern recognition, with an increasing number of researchers using new platforms and CNNs for forest research [22–24]. Compared with that of traditional methods, the main advantage of CNNs is their ability to automatically learn the features of imagery through deep networks without excessive manual feature extraction.

There have been an increasing number of studies on tree species identification using visible and multispectral cameras mounted on UAV platforms [19]. A visible camera can be used to obtain ultra-high spatial resolution red-green-blue (RGB) imagery data at a low cost, and these data do not require professional calibration and data pre-processing [22,25]. These advantages have led many researchers to apply RGB imagery to studies of tree species classification [14,20,26]. Multispectral cameras can obtain the spectral response specific frequencies across the electromagnetic spectrum including in the infrared band. The spectral information in the infrared band is related to the structure and morphology of leaves/canopies [27] and the biochemical composition of trees [8,28]. Therefore, multispectral imagery is used not only for tree species classification but also for inversion of other parameters of the forest [29–31].

Although multispectral and RGB imageries have their own advantages in terms of acquisition cost, spatial details, and wavelength information, few studies have been conducted for comparing their performance [32,33]. One study found that multispectral imagery had higher classification accuracy compared to RGB imagery to identify marine macroalgae in February [34]. However, another study showed that the RGB imagery acquired using drones was more important than multispectral cameras for mapping invasive species in June [35]. Apart from the disagreement, in most of these studies, the experi-

mental sites and study areas have encompassed urban forests and plantations [9,19], not considering the complexity of the forest ecosystem.

In this study, we selected the rarely trafficked, highland alpine ecological environment of the Wanglang Nature Reserve as study area. The main questions of our study are: (1) When was the most suitable time to identify Faxon fir for both multispectral and RGB imagery? (2) Was the accuracy of multispectral imagery better than RGB for different months? We combined RGB and multispectral imagery throughout the growing season with a semantic segmentation method (Deeplabv3) to map Faxon fir. We also compared models perform of multispectral imagery with RGB imagery and tested the effects of added vegetation indices (VIs) to the models.

## 2. Materials and Methods

### 2.1. Study Area

The study area is located in the Wanglang National Nature Reserve, Pingwu County, Sichuan Province, China (103°55′–104°10′E, 32°49′–33°02′N) (Figure 1). The Wanglang Nature Reserve is situated at the intersection of the Qinghai–Tibetan Plateau and the Sichuan Basin and is in the western Sichuan alpine valley region at the northern edge of the Hengduan Mountains. It has one of the best-preserved primary forests in northwest Sichuan [36,37]. The reserve has a semi-humid climate of the Danba–Songpan, with an altitude of 2300–4980 m. The climate has pronounced vertical zonality, with an average annual temperature of 2.5–2.9 °C, a minimum temperature of −17.8 °C, a maximum temperature of 26.2 °C, and an annual rainfall of 862.5 mm [38]. At this site, Faxon fir was the dominant species; *Picea purpurea*, as the main companion species, was less abundant [1,39].

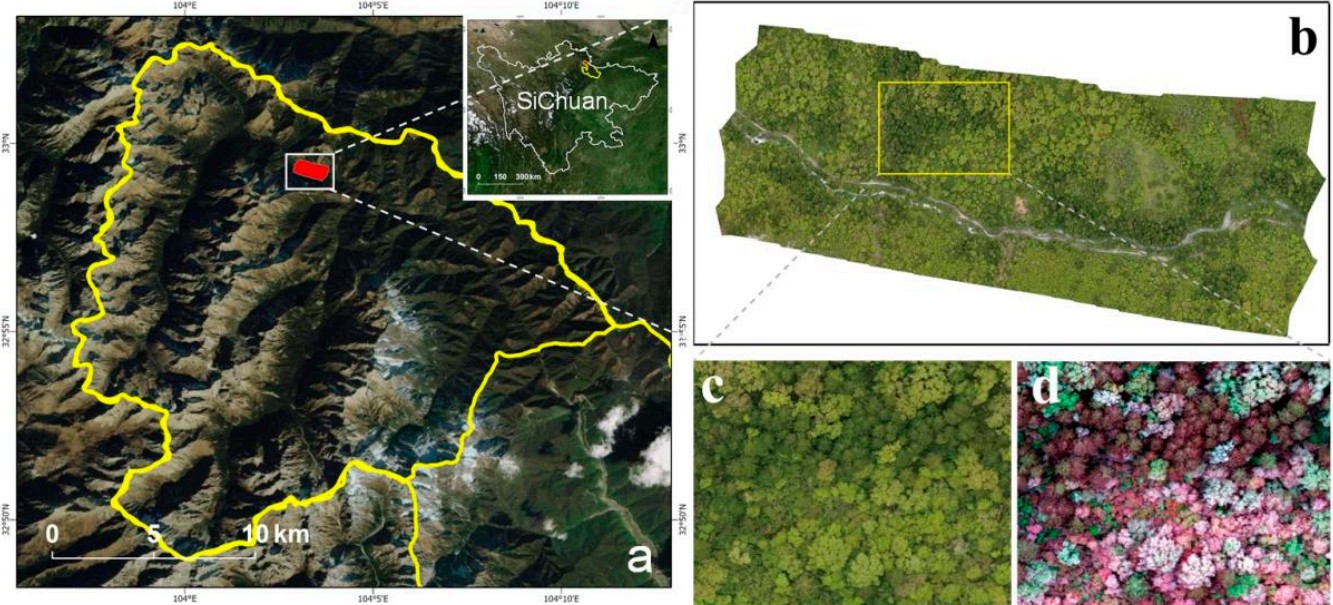

**Figure 1.** Overview of the study area: (**a**) satellite Image of the study area; (**b**) June data collection in the study area; (**c**) a subarea of Red–green–blue (RGB) imagery in June; and (**d**) a false color composite (green-red-near infrared) of the subarea of the multispectral imagery in June.

### 2.2. Data Acquisition

On each UAV flight, a DJI 4 Multispectral (P4M) (SZ DJI Technology Co., Ltd., Shenzhen, China) camera was mounted. The P4M camera was fitted with six 1/2.9-inch complementary metal oxide semiconductor (CMOS) sensors, including one color sensor for visible imaging and five monochrome sensors for multispectral imaging. These were used to measure spectral response in the blue, green, red, red-edge, and Nir bands (Table 1). Each sensor provided 2.08 million effective pixels (2.12 million total pixels). Photographs

were captured to ensure an 80% frontal overlap and a 70% side overlap, as needed, to meet the requirements of the completed map without voids and blind areas (Table 2).

**Table 1.** Drone equipment parameters.

| Image Acquisition Equipment | DJI P4M |
|---|---|
| Image sensor | 1/2.9-inch CMOS |
| Blue bands | 450 ± 16 nm |
| Green bands | 560 ± 16 nm |
| Red bands | 650 ± 16 nm |
| Red edge band | 730 ± 16 nm |
| Near-infrared band | 840 ± 26 nm |
| Acquisition Mode | Snapshot |
| Optics | f/2.2 |
| FOV | 62.7° |

**Table 2.** Flight parameters and image information from the UAV on different dates.

| Flight Date | Flight Time | Flight Height | Total Images | Spatial Resolution (m) | Flight Area (ha) |
|---|---|---|---|---|---|
| April 21 | 14:23 | 500 m | 223 × 6 | 0.21 | 194.24 |
| May 23 | 14:31 | 300 m | 121 × 6 | 0.13 | 40.08 |
| June 17 | 13:07 | 400 m | 145 × 6 | 0.17 | 101.80 |
| August 27 | 16:07 | 400 m | 145 × 6 | 0.17 | 99.19 |
| September 27 | 16:51 | 400 m | 145 × 6 | 0.17 | 99.04 |
| October 27 | 14:17 | 400 m | 145 × 6 | 0.17 | 94.47 |

UAV imagery was collected during the growing season 2021 between early spring and late autumn 2021, including six total flights on April 21, May 23, June 17, August 27, September 27, and October 27 at the Wanglang Mountain Remote Sensing Observation and Research Station of Sichuan Province (Table 2). The DJI P4M UAV has one color sensor for visible imaging and five monochrome sensors for multispectral imaging, so each shot will generate one visible photo and five photos in different wavelengths. The UAV images for all dates were processed in DJI Terra V3.1 (SZ DJI Technology Co., Ltd., Shenzhen, China), resulting in a total of 6 RGB and 6 multispectral orthophoto imageries with the same resolution (Table 2). In addition, for the visual inspection of the study's results, we also used the same flight parameters to acquire UAV images at an adjacent site at the same time.

Training deep learning models requires accurate data with large sample sizes. Owing to the high altitude of the study area, the growing season of the vegetation commenced relatively late in the year. Figure 2 shows that the broadleaf trees had not gotten their leaves in April, and the Faxon fir could be explicitly identified in the image. Therefore, we selected the April imagery for visual interpretation and manual delineation for derivation of the masks using ArcGIS v.10.2.1 (ESRI, Redlands, CA, USA). For masks for the other months, we modified the masks based on the April mask to ensure data accuracy.

Considering the effect of spatial resolution on model accuracy and ensuring maximum preservation of the original image information, we resampled the April and May imagery to 0.17 m using nearest neighbor method and aligned the April image extent with that of June. There is no sample signal in the sample site, so the DJI P4M UAV could not use the RTK positioning function during the flight, and could only rely on the GNSS positioning function, so it would lead to a horizontal error of ±1.5 m in the images of different months. Since the offset was small, we aligned the images through spatial correction. Given the operating environment of the deep learning models in this study, so we cropped orthophoto imageries to 256 × 256 pixels before training the CNN. The orthophoto imagery was cropped 848 images in May and 1326 images in other months. We randomly selected 90% of the images as training data and 10% as validation data. For better model generalization,

we performed data augmentation during model training. Finally, we also used adjacent sample plots to validate the trained model.

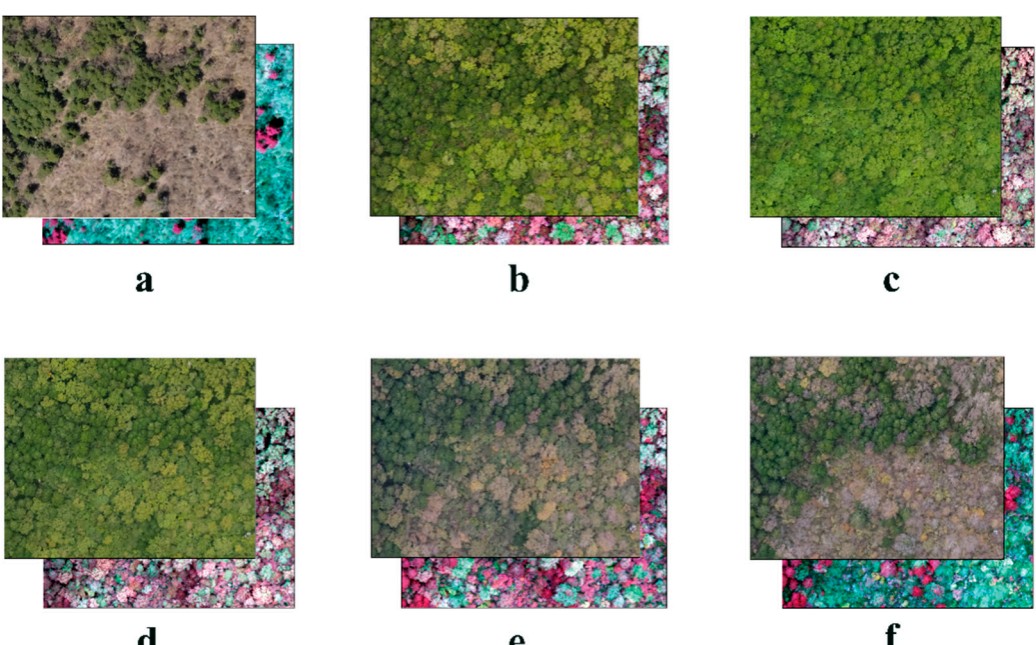

**Figure 2.** RGB and multispectral imagery collected by drones for different months: (**a**) April 21, (**b**) May 23, (**c**) June 17, (**d**) August 27, (**e**) September 27, and (**f**) October 27.

### 2.3. Vegetation Indices (VIs)

VIs are often used within remote sensing classification [40–42]. They represent the vegetation condition using a simple mathematical combination or transformation of two or more bands while maximizing the elimination of other contributing factors, such as the soil background, atmosphere, and solar target-sensor geometry [43,44]. The addition of VIs to deep learning models can enhance the vegetation status information, which improves the potential of deep learning in the field of remote sensing [45–47].

From the DJI P4M, broad-band ratio VIs that eliminate the background effect are available for vegetation canopies in the UAV images directly without radiation correction [48–50]. Therefore, we have selected the three most widely used VI ratios for the RGB and multispectral imagery, respectively (Table 3).

(1)    Multispectral VIs

Normalized difference vegetation index (NDVI) is currently the most widely used vegetation index in the world. NDVI can partially eliminate the influence of radiation variation related to solar altitude angle, satellite observation angle, topography, cloud shadow, and other atmospheric conditions; its result is limited between [−1,1], which avoids the inconvenience of using too large or too small data; it is also the best indicator of vegetation growth status and vegetation cover [51,52]. Red-edge NDVI is a vegetation index that uses red-edge bands to estimate the health of vegetation. It measures NIR light penetrating into the lower part of the canopy and is therefore well suited to analyze the health and vigor of mid- to late-stage crops [53,54]. The green normalized difference vegetation index (GNDVI) is more sensitive to changes in vegetation chlorophyll content than NDVI [55]. It combines spectral values in the green and near-infrared bands, and the index is used to detect wilted or aged crops and measure nitrogen content in leaves, as well as to monitor vegetation in dense canopy or mature stages.

(2)    RGB VIs

The normalized green-red difference index (NGRDI) enables us to assess the health of vegetation by measuring the difference between red and green light [56,57] and is also an

effective index to distinguish vegetation from non-vegetation. The normalized green–blue difference index (NGBDI) can assess the health of vegetation by measuring the difference between the green and blue bands. When vegetation is healthy, they absorb more blue light and therefore have lower NGBDI values [58]. The visible-band difference vegetation index (VDVI) combines the reflectance of vegetation in the green light band and absorption in the red and blue light bands in a form similar to NDVI with values in the range of [−1,1], which can better distinguish vegetation from non-vegetation [59,60].

**Table 3.** Vegetation indices selected for data processing.

| Image | VI Name | Abbreviation | Equation and Derivation |
|---|---|---|---|
| Multispectral imagery | Normalized difference vegetation index [61–63] | NDVI | $(NIR - Red)/(NIR + Red)$ |
| | Red-edge NDVI [64] | NDVIre | $(NIR - RE)/(NIR + RE)$ |
| | Green normalized difference vegetation index [65] | GNDVI | $(NIR - Green)/(NIR + Green)$ |
| RGB imagery | Normalized green red difference Index [61] | NGRDI | $(Green - Red)/(Green + Red)$ |
| | Normalized green blue difference index [66] | NGBDI | $(Green - Blue)/(Green + Blue)$ |
| | Visible-band Difference Vegetation Index [59] | VDVI | $(2 \times Green - Blue - Red)/(2 \times Green + Blue + Red)$ |

## 2.4. Classification Method

DeepLabv3 was specifically designed to address semantic segmentation [67]. Compared with previous DeepLab series models, DeepLabV3 increases the receptive field exponentially without reducing or losing the spatial dimension and improves the performance of segmentation tasks. DeepLabv3 (Figure 3) uses Resnet 34 [68] as the backbone network to extract features and classify each pixel. This is undertaken using the improved Atrous Spatial Pyramid Pooling (ASPP) module by including batch normalization and image-level features. In the ASPP network, on top of the feature map extracted from the backbone, one $1 \times 1$ convolution and three $3 \times 3$ convolutions with rates = (12, 24, 36) are applied to handle segmenting the object at different scales. Then, image-level features are used to incorporate global context information by applying global average pooling. The output from the ASPP network was passed through a $1 \times 1$ convolution to obtain the actual size of the image. This was then used as the final segmented mask for the image. This study used a deep learning framework based on Pytorch. Moreover, we chose stochastic gradient descent (SGD) as our optimizer with a momentum strategy and the number of training epoch was 100. If the loss does not drop within 5 epochs, the training stops running.

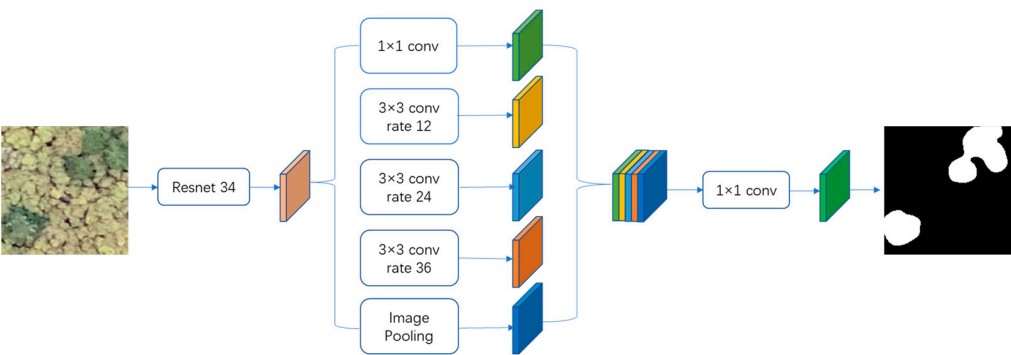

**Figure 3.** Architecture of DeepLabv3 for tree species segmentation.

*2.5. Classification Accuracy Assessment*

In this study, we evaluated the accuracy of the model based on the precision (P), recall (R), F1 score (F1), and intersection over union (IoU). The formula is defined as follows:

$$Precision = \frac{TP}{TP+FP} \tag{1}$$

$$Recall = \frac{TP}{TP+FN} \tag{2}$$

$$F1 = \frac{2 \times P \times R}{P+R} \tag{3}$$

$$IoU = \frac{TP}{FN+FP+TP} \tag{4}$$

*TP* refers to true positive samples, that is, predicting positive samples as positive samples. *FP* refers to false positive samples, that is, predicting negative samples as positive samples. *FN* refers to false negative samples, that is, predicting positive samples as negative samples.

**3. Results**

*3.1. Model Accuracy in Different Months*

The classification models trained using UAV imagery and CNNs achieved effective classification results in different months (Table 4 and Figures 4 and 5). The RGB model had the highest accuracy for April, with an F1 of 91.06% and IoU of 83.72%, with the accuracy decreasing slightly in September and October. However, the accuracy for May (F1 = 89.12%, IoU = 80.21%) and August (F1 = 87.98 %, IoU = 78.52 %) models decreased slightly. The accuracy of the RGB model was lowest in June, with an F1 of 86.91% and an IoU of 76.81%. The model accuracy for the RGB imagery varied considerably across months, with the F1 varying by nearly 5% and the IoU varying by nearly 7%.

**Table 4.** Faxon fir mapping model accuracies in different months. For the multispectral (MS) and RGB imagery, the highest intersection over union (IoU) and F1 score (F1) are highlighted in bold.

| Month | April 21 | | May 23 | | June 17 | | August 27 | | September 27 | | October 27 | |
|---|---|---|---|---|---|---|---|---|---|---|---|---|
| Input Data | MS | RGB | MS | RGB | MS | RGB | MS | RGB | MS | RGB | MS | RGB |
| Precision | 90.45% | 91.24% | 89.77% | 88.21% | 89.82% | 87.75% | 90.06% | 88.27% | 91.04% | 90.40% | 90.06% | 90.68% |
| Recall | 90.24% | 90.87% | 91.20% | 90.06% | 89.38% | 86.08% | 89.85% | 87.70% | 91.25% | 90.65% | 89.63% | 90.35% |
| F1 | 90.34% | **91.06%** | 90.48% | 89.12% | 89.60% | 86.91% | 89.95% | 87.98% | **91.15%** | 90.52% | 89.84% | 90.52% |
| IoU | 82.50% | **83.72%** | 82.56% | 80.21% | 82.31% | 76.81% | 81.69% | 78.52% | **84.82%** | 82.66% | 81.56% | 82.67% |

The multispectral model had higher accuracy throughout the acquisition dates, with the highest accuracy in September with an F1 of 91.15% and IoU of 84.82%. In April, May, and June, the accuracy of the model decreased slightly (F1 > 90%, IoU > 82%). In August, the model accuracy was lower (F1 = 89.84%, IoU = 81.56%). The lowest accuracy of the multispectral model was in October, with an F1 of 89.84% and IoU of 81.56%. The model accuracy from the multispectral imagery in different months differed slightly, with F1 and IoU being approximately 2%.

To show vegetation variability [69], we calculated the standard deviations (STD) of RGB imagery for the six months. The STD in Figure 6 was measured by averaging the STD of all pixels in the three visible bands, where a higher STD indicates a higher variability of the feature. We found that the variability in May, June, and August was less than that that in April, September, and October (Figure 6).

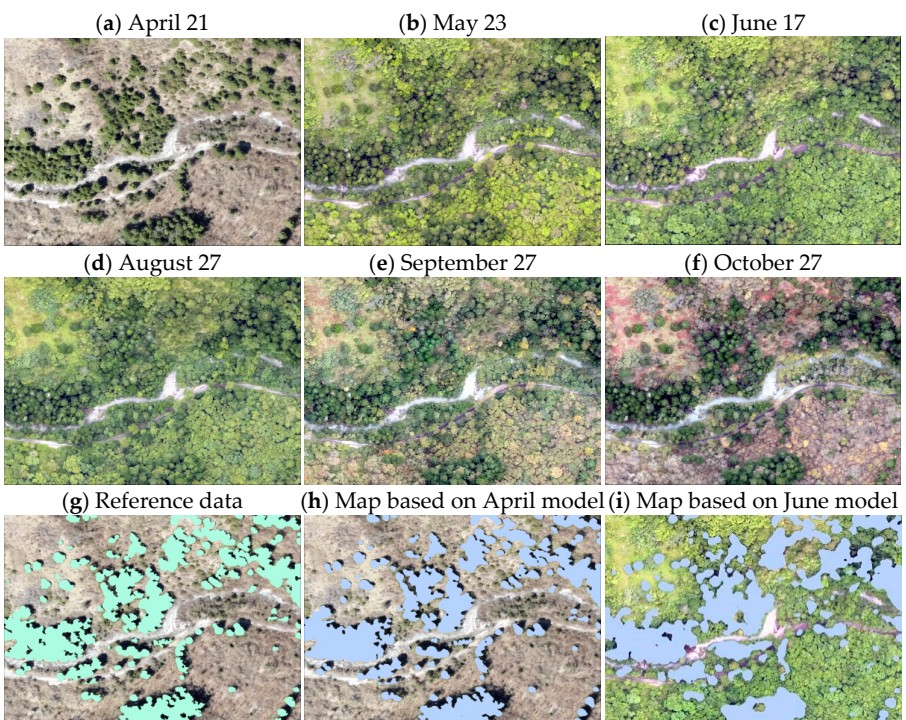

**Figure 4.** UAV-based RGB orthomosaics for different months and the associated Faxon fir identification map: (**a**) April 21, (**b**) May 23, (**c**) June 17, (**d**) August 27, (**e**) September 27, (**f**) and October 27 (**g**) manually labeled reference data in April; (**h**) identification map of RGB imagery on April model (highest accuracy), and (**i**) identification map of RGB imagery on June model (lowest accuracy).

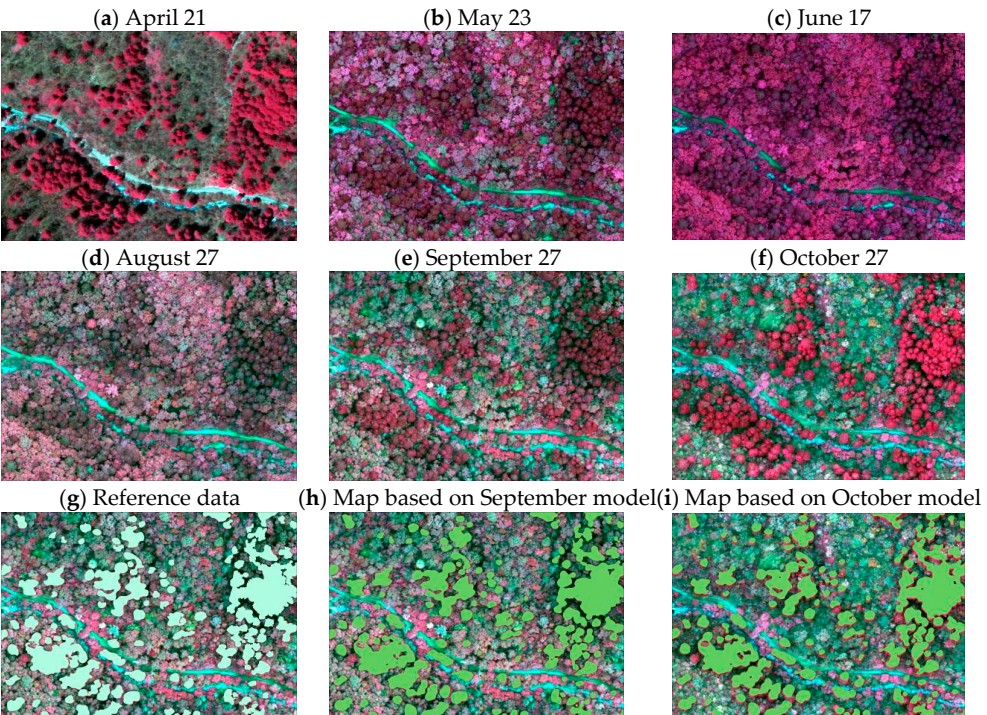

**Figure 5.** UAV-based multispectral orthomosaics in different months and the associated Faxon fir identification map: (**a**) April 21, (**b**) May 23, (**c**) June 17, (**d**) August 27, (**e**) September 27, (**f**) October 27, (**g**) manually labeled reference data in September; (**h**) identification map of MS imagery on September model (highest accuracy), and (**i**) identification map of MS imagery on October model (lowest accuracy).

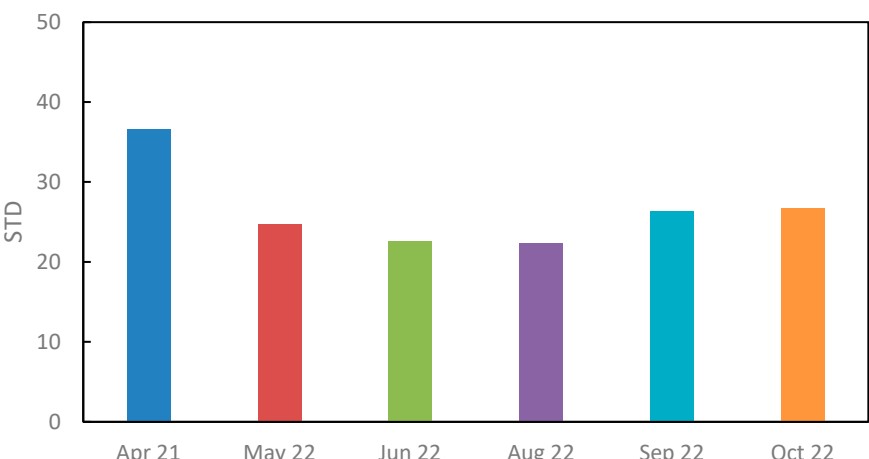

**Figure 6.** Variability in UAV orthomosaics for April, May, June, August, September, and October.

### 3.2. Comparison of the Multispectral and RGB Models

The accuracy of the multispectral model was higher than that of the RGB model for May, June, August, and September (Figure 7). In May and September, the accuracy of the multispectral model was slightly higher than that of the RGB model. In August, this difference was more pronounced, given that the F1 of the multispectral model was 1.97% higher and the IoU was 3.17% higher than those of the RGB model. In June, this difference reached its highest value, with the F1 of the multispectral model being higher by 2.69% and the IoU being 5.5% higher than those of the RGB model (Figure 8). However, in April and October, this situation was reversed. The RGB model was slightly more accurate than the multispectral image model, wherein the F1 values of the RGB model were 0.71% and 0.67% higher and the IoU was 1.23% and 1.11% higher, respectively (Figure 8).

To further validate our model results, we also used UAV images from June and October from the adjacent sample sites for model classification (Figure 9). Like the results from the original sample site (Figure 8), the identification ability of the multispectral model for the adjacent sample site was higher than that of the RGB model in June; however, in October, the identification ability of the multispectral and RGB model was close.

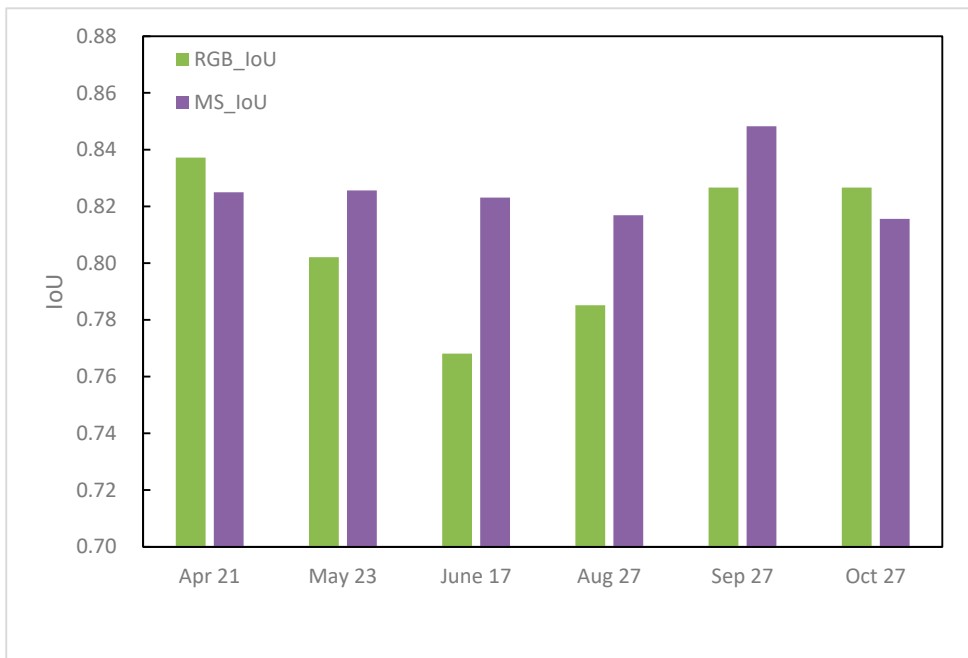

**Figure 7.** Comparison of the IoU for the RGB and MS models for different months.

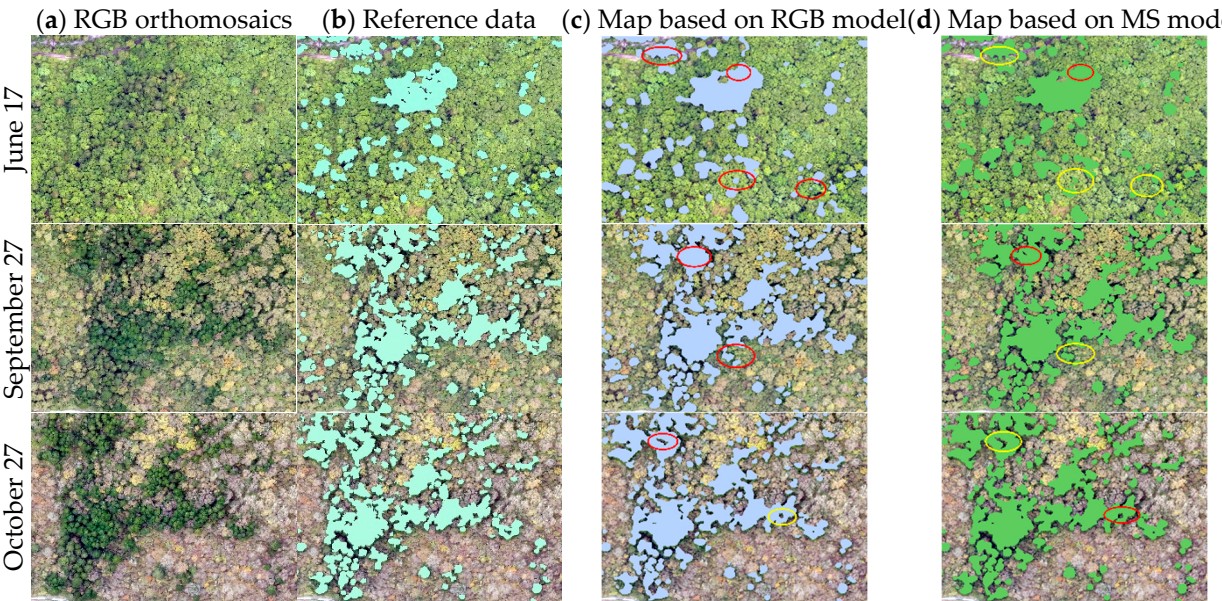

**Figure 8.** Faxon fir identification for different months. (**a**) UAV-based RGB orthomosaics, (**b**) manually labeled reference data, (**c**) identification map based on the RGB model, (**d**) identification map based on the MS model. The yellow and red circles represent correct and incorrect identification results, respectively.

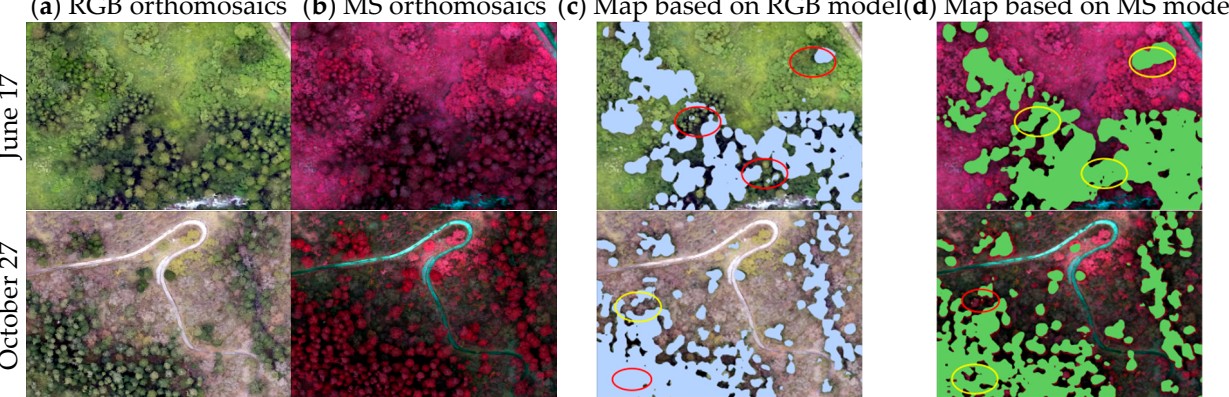

**Figure 9.** Faxon fir identification of trained CNN on the adjacent sample sites for different months. (**a**) UAV-based RGB orthomosaics, (**b**) UAV-based MS orthomosaics, (**c**) identification map based on the RGB model, (**d**) identification map based on the MS model. The yellow and red circles represent correct and incorrect identification results, respectively.

*3.3. Model Accuracy with Added VIs*

Compared with the imagery model without the addition of VIs, the RGB model with added VIs showed an improvement in F1 by approximately 1%; IoU increased by approximately 2% in May and June (Table 5). This was followed by the results from August, with F1 improving by 1.41% and IoU improving by 0.89%. In April, September, and October, the model accuracy increased or decreased slightly, but the overall change was not significant. The multispectral model with added VIs showed little overall change in F1. However, the IoU decreased by approximately 1% in June and September and slightly increased or decreased in the other months. Although the RGB model showed some improvement in accuracy with the addition of VIs in May and June, the accuracy was still lower than that of the multispectral model for most months without the addition of VIs (Figure 10).

**Table 5.** Accuracy of the model with added vegetation indices (VIs) in different months. For MS and RGB imagery, the highest IoU and F1 are highlighted in bold.

| Month | April 21 | | May 23 | | June 17 | | August 27 | | September 27 | | October 27 | |
|---|---|---|---|---|---|---|---|---|---|---|---|---|
| Input Data | MS_VI | RGB_VI | MS_VI | RGB_VI | MS_VI | RGB_VI | MS_VI | RGB_VI | MS_VI | RGB_VI | MS_VI | RGB_VI |
| Precision | 90.26% | 91.33% | 90.02% | 89.50% | 89.61% | 88.95% | 89.78% | 89.43% | 91.09% | 90.50% | 90.13% | 90.54% |
| Recall | 90.32% | 91.50% | 91.55% | 91.10% | 89.86% | 87.32% | 89.39% | 88.33% | 91.02% | 90.29% | 89.84% | 90.00% |
| F1 | 90.29% | **91.41%** | 90.78% | 90.29% | 89.73% | 88.13% | 89.59% | 88.88% | **91.05%** | 90.40% | 89.99% | 90.27% |
| IoU | 82.36% | **84.33%** | 83.06% | 82.16% | 81.27% | 78.61% | 81.11% | 79.93% | **83.57%** | 82.37% | 81.79% | 82.24% |

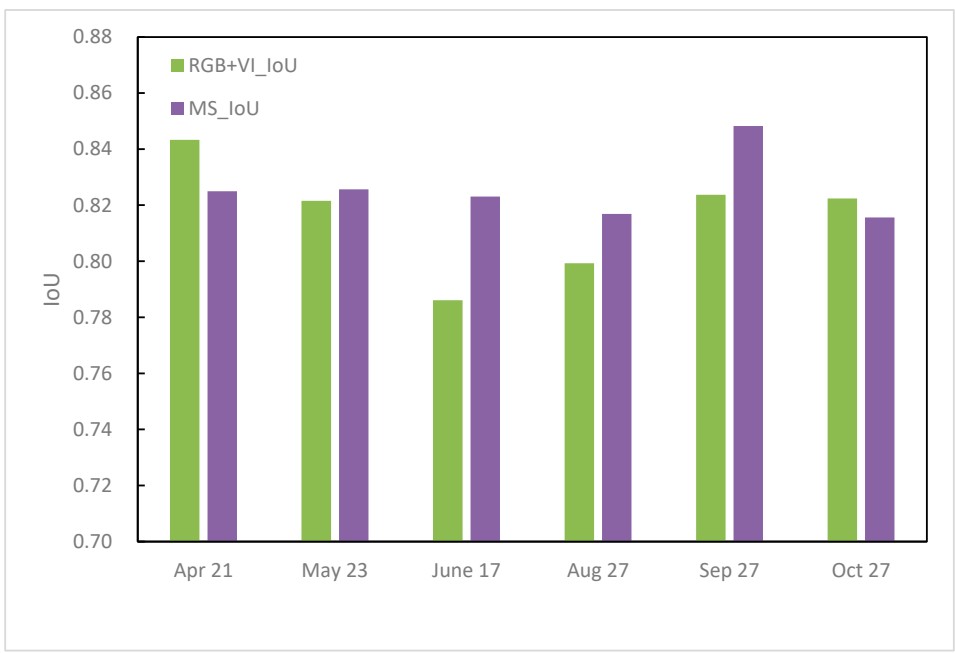

**Figure 10.** Comparison of the IoU of the RGB model with added VIs and MS imagery without added VIs for different months.

## 4. Discussion

### 4.1. Model Performance in Different Months

Although the acquisition dates of the UAV imagery varied among the models, the training models constructed in this study using CNNs with UAV imagery had high levels of overall accuracy (F1 > 86% and IoU > 76%). This meant that they were able to accurately identify Faxon fir throughout the growing season. This indicates that using the end-to-end learning capabilities of CNNs can fully facilitate the exploitation of high resolution of UAV imagery [14]. They can also be used to obtain accurate tree species distribution, which is conducive to improving the efficiency of forestry and ecological conservation work.

The accuracy of the RGB model varied considerably in different months, whereas the multispectral model had consistently high accuracy (Table 4). Although RGB imagery can be used to map species distribution accurately [70–72], its level of accuracy is relatively low in periods with less interspecies variation. This is particularly true for tree species in the same family and genus because they predominantly have similar structures and textures [73]. Schiefer et al. [14] found that it was difficult to effectively distinguish between two similar tree species, namely *Quercus petraea* and *Quercus robur*, using only high-resolution UAV-based RGB imagery. Multispectral imagery is more strongly affected by differences between species because it can obtain red-edge and NIR information and capture subtle differences in vegetation [74,75].

The highest model accuracy for the entire observation period was recorded in September. In contrast with our expectations, the accuracy of the imagery model was not highest during April and October, when trees showed the most phenological variation [32]. This may be because this period showed a balance between inter- and intra-species spectral variations [76]. This suggests that when using UAVs to collect remote sensing images, we should not only focus on the phenological information for the vegetation but also pay attention to intra-species variation.

### 4.2. Differences in the Capabilities of Multispectral and RGB Imagery

For the growing season, the accuracy of the multispectral model was higher than that of the RGB model. During periods of low interspecific variation (Figure 6), such as in May or June, the gap between the RGB and multispectral models was substantially larger (Figure 5). Ahmed et al. [77] found that the overall accuracy of multispectral imagery was up to about 10% higher than that of RGB imagery when using UAV imagery in June for land cover classification. Tait et al. [34] showed that the accuracy of multispectral imagery was 2% higher than that of RGB imagery in February for habitat classification. These results suggest that the capability of multispectral imagery is superior to RGB imagery in times of lower interspecies differences.

However, in October and April, the model accuracy for the RGB imagery was slightly higher than that of multispectral imagery (Figure 5). In cases of large interspecies variation, only RGB imagery can be used to accurately map species distribution based on canopy color and texture [70,71]. The photosynthesis of vegetation was not strong during these periods [78], resulting in a reduced ability to capture vegetation responses in the infrared and near-red bands of multispectral imagery. Weisberg et al. [35] found that during key phenological periods, acquiring RGB imagery was more important than acquiring multispectral imagery for invasive plant identification in the Great Basin of the western United States. This is in line with the results of our model for October and April. In terms of practical applications, ultra-high-resolution RGB UAV imagery can be easily acquired at a relatively low cost. Many studies have also demonstrated that the increase in spatial resolution can improve identification accuracy [14,79]. These results further highlight the considerable potential of RGB imagery in forestry applications.

In contrast with satellite remote sensing, which can automatically capture images without interruption, UAV remote sensing requires researchers to make their own flight plans and select suitable flight loads [80]. Our study demonstrated that the superiority of multispectral imagery over RGB imagery gradually decreased or even disappeared as the difference between vegetation species gradually increased. Therefore, to map tree species accurately, we can choose a period with substantial interspecific differences between plant species to acquire high-resolution RGB drone images. If the period is missed, we can also obtain multispectral or even hyperspectral data by carrying loads with more wavelengths.

Although our application was carried out on a large spatial extent, certain limitations still exist. As our study area is located in a national nature reserve and there is no network signal in the sample site, acquiring the real distribution range of different tree species is very difficult. In this study, we only determined the distribution of a single tree species through multi-seasonal images. Therefore, in the upcoming work, we would like to add broadleaf species in our next work to further study the relationship between tree species identification and UAV monitoring. Meanwhile, canopy height models can be introduced to improve the classification ability of the model when studying multi-species distribution.

### 4.3. Potential of Vegetation Indices in Tree Species Classification Using CNN

The addition of VIs to the RGB model accuracy increased significantly in May, June, and August. This is consistent with previous study findings that adding VIs to CNN models can improve the classification performance [46,47,81]. This suggests that vegetation indices with simple computations and stable features can enhance vegetation information and complement deep learning based on deep feature extraction. However, in April,

September, and October, the VIs was less useful for the RGB model. This means that VIs may not be able to further highlight the existing interspecific differences during periods with substantial interspecific differences. The RGB model with added VIs was still less effective than the multispectral model without added VIs, except in April and October. This finding highlights the importance of NIR and IR information in multispectral imagery.

Adding VIs to the multispectral model led to no changes in F1, and only in September and June did the accuracy of IoU decrease by approximately 1%. After adding three VIs, the input data of the multispectral model had eight bands, and these bands had some correlation. Therefore, it is not effective to improve the model accuracy and may cause some overfitting of models. Kerkech et al. [47] also found that the addition of too many vegetation indices did not improve the model accuracy, rather the accuracy decreased slightly.

We only selected the most common specific vegetation indices such as NDVI, which often have some level of oversaturation in dense forests and cannot effectively reflect vegetation information in summer [42]. If we choose vegetation indices from other remote sensing products that can better reflect the actual photosynthesis of vegetation in future studies, such as sun/solar-induced chlorophyll fluorescence [82], the accuracy of the model would likely increase substantially. In conclusion, the vegetation index for enhancing vegetation information can be regarded as a specific type of data enhancement in deep learning, and we need to further explore the potential of combining VIs with CNN.

## 5. Conclusions

We monitored the study area monthly and acquired UAV imagery data from the sample sites for the entire growing season. Our study showed that UAV imagery combined with CNNs could accurately map the distribution of Faxon fir throughout the growing season in a subalpine plateau environment. We found that although there were no pronounced phenological differences in Faxon fir, the acquisition time of the UAV images still had a considerable impact on the classification accuracy. Upon comparing RGB and multispectral images from different months, our results showed the optimized time of the year for identifying Faxon fir using UAV imagery is during the peak of the growing season when having a multispectral imagery, but when only and RGB imagery is available off growing season is better. The additional VIs only significantly improved the accuracy of the RGB model in summer and were still less effective than the multispectral model without the addition of VIs. Our study highlights the great potential of UAV platforms and can serve as a guide to optimize UAV observation plans for the application community, facilitating the application of UAV in forestry and long-term ecological monitoring.

**Author Contributions:** W.S.: conceptualization, methodology, formal analysis, writing—original draft, writing—review and editing. X.L.: conceptualization, writing—review and editing, funding acquisition. J.S.: writing—review and editing, funding acquisition. Z.Z.: investigation, data curation. S.W.: conceptualization, writing—review and editing, funding acquisition. D.W.: writing—review and editing. W.Q.: writing—review and editing. H.H.: visualization. H.Y. (Huping Ye): writing—review and editing. H.Y. (Huanyin Yue): writing—review and editing, funding acquisition. T.T.: writing—review and editing, funding acquisition. All authors have read and agreed to the published version of the manuscript.

**Funding:** This research was supported by the Strategic Priority Research Program of the Chinese Academy of Sciences (No. XDA19050501, No. XDA19040104), the Scientific Research Foundation of China University of Geosciences (Wuhan) (No. 162301192642) and the National Natural Science Foundation of China (42001314). Torbern Tagesson was additionally funded by the Swedish National Space Agency (SNSA 2021-00144) and FORMAS (Dnr. 2021-00644).

**Data Availability Statement:** The data presented in this study are available from the corresponding author upon reasonable request.

**Acknowledgments:** The authors are thankful to the staff who acquired and managed the UAV imagery by Wanglang Mountain Remote Sensing Observation and Research Station of Sichuan Province. We also thank Jing-Hua Chen and student Chen Zheng for their comments on the paper.

**Conflicts of Interest:** The authors declare that have no known competing financial interests or personal relationships that could have appeared to influence the work reported in this paper.

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
