# Peer review of "Optimizing Observation Plans for Identifying Faxon Fir (Abies fargesii var. Faxoniana) Using Monthly Unmanned Aerial Vehicle Imagery"

_remotesensing, doi:10.3390/rs15082205_

Round 1

Reviewer 1 Report

The paper describes how UAS systems can be used to monitor forests in natural reserves, and the authors used RGB and NIR images acquired by UAV to detect specific tree species. Unpiloted aerial vehicles are more and more used for forest monitoring. In addition to the forest mapping (aerial extent), there is a lot of interest in comparison of UAV data with LiDar data, and the use of UAV data for forest biomass and structure. The authors could strengthen their work by referring to these other developments in UAV forest monitoring.

Also, the manuscript is rather short on the methodological aspects, and several aspects (like uncertainty on image stitching etc) could be further developed in the text with additional tables/figures.

General comments

1-In the introduction, you are very positive about the use of UAV for forest monitoring. While this field is rapidly evolving, with a lot of good applications, there are also some aspects of traditional forest monitoring that cannot be surveyed with UAV data. LiDAR is often cited to resolve a few of these issues, notably the issue of underdetection of vegetation undergrowth and estimation of forest biomass. I would recommend to include these aspects in the introduction to have a more nuanced presentation of the possibilities of UAV forest monitoring. For an overview, you could refer to classical work by Dandois et al (2015), Iglhaut et al (2019) or more recent work by e.g. Apostol et al. (2020) or Gobbi et al (2022).  

2-The research question is not entirely clear. Can you rewrite L104-105 (reads more as objective) as a research question?

3-More information is needed on the flights: can you add information on the flight velocity, height, ground sample distance? This can be added to Table 1. See e.g. Table 1 in Iglhaut et al (2019) or Table 2 in Gobbi et al (2020).

On L151-152, you mention that resampling of images was necessary but it is not clear why this is the case as we do not have any information on the resolution of the images, the pixel size or ground sampling distance. Can you add this information in a table?

4-The classification methods are explained in great detail, which is helpful to understand the results. Other methods need to be further developed. It is unclear to me how the images were processed and stitched? Are all images stitched together for the VI products? What was the uncertainty on the geolocation of the images? What is the horizontal absolute error on the intermediate products, and how do you deal with comparing maps with a certain error in the geolocation? What is the uncertainty of the final product?

5-What about independent validation of the results? The final products are now evaluated based on digitalisation by the authors on the images that are used for the forest monitoring. Somehow, they are not giving an independent dataset (bias by the source of the data, and by the user). Is there the possibility to verify with forest inventory data for the region? Are there e.g. forest inventory plots that can serve as basis for the comparison? See comment in Iglhaut et al (2019)

6-Is there an added value of looking at the point clouds, and extract data on forest structure from the canopy height models? This might be interesting to evaluate forest state, and to have a first idea of differences in forest stages based on tree height, canopy density, etc.

Detailed comments
L39-40: Can you rewrite the sentence? Not entirely clear to me
L42: Can you check the use of the word "practicability"?
L65-66: Please check wording: how can the acquisition of images be a "technical support"?
L68: I agree that you have a lot of spatial information with UAV, but I do not agree that you have a lot of "spectral" information when using RGB UAV.
L69: check wording : "lso increases inter- and intra-class variability 69
in features"
L116-118: Given the large altitudinal range, I would expect to have large spatial variability in temperature and also rainfall. Are the values for the entire area? Can you specify where the meteo data come from (source of the data, and interval of measurement)?
L170: Can you specify why you used these VIs? Is there a specific reason for selecting these VIs here?
L228/Figure 6: The analysis of the variance in the data is interesting, and could be furthe developed. How did you measure the variance? Is this the sample variance in the reflectance values (RGB)? For RGB, you would have the variance in three bands, but you report one value as "STD". How do I need to read Figure 6? Is this the variance, the standard deviation of the reflectances in one of the bands, or is this about variability? Can you check and correct?
Figure 6: It would be interesting to see the mean + variance for the different dates, and the different spectral bands.
L234-244: Can you check the values that are reported here. If you compare with a given reference (let say, 89.60% and 86.91%), a change of -3% compared to 89.60% would be (89.60% - 3% of 89.60). Please check, and verify also in the next paragraphs.
L337: not entirely clear how the network signal is affecting the results. It probably has an important effect on your accuracy, or have you corrected the images with PPK-GLONAS systems?
L342: Can you be more specific about "more study areas and tree species"? What are the interesting areas that one needs to target + why? What are the key species?

"UAV" => To use gender-neutral language, people often prefer to use "unpiloted aerial vehicles" rather than "unmanned"

Figure 1: It is a bit difficult to know the geographic location with this satellite image. Can you add latlong coordinates in the figure? And then also add a topographic map (note that you refer to a topographic map in the caption, but I rather see a satellite image (?)). Can you check and correct? Can you add the scalebar to inset b, c, and d?
Table 1: Can you provide additional data- see comment above
Table 2: Not entirely clear to me why all image quantities are "*5"? Can you explain? Also, can you indicate what is meant with the "spatial resolution"? Is this the ground sample distance? Please check the articles mentioned below for standard annotation of variables of UAV surveys.
Figure 4: In the delineation of the forest species (reference data), how did you deal with shadow effects? For the April 21 data, there seems to be quite a bit of shadow present SE of the trees. How was this accounted for?
Figure 4-5-8-9: Can you add scalebar?

References
Apostol et al (2020). doi:10.1016/j.scitotenv.2019.134074
Dandois et al (2015). doi:10.3390/rs71013895
Gobbi et al (2022). https://doi.org/10.1016/j.foreco.2022.120554
Iglhaut et al (2019). https://doi.org/10.1007/s40725-019-00094-3

Reviewer 2 Report

Comments to authors

This article describes a research study that combined UAV imagery and CNNs to identify Faxon fir and explored the identification capabilities of multispectral (five bands) and red-green-blue (RGB) imagery under different months, which can provide guidance for optimizing observation plans regarding data collection time and UAV loads, and could further help enhance the practicability of UAVs in forestry and ecological research. Therefore, it has certain research value. However, there are some issues as follows:

1. The training sample in this article was completed through visual interpretation, please explain its reliability.

2. Please introduce the optimizer and other parameters of the deep learning model.

3. What reasons did the author consider when selecting the vegetation index?

Besides, there are some detail issues, such as:

1. There is a mistake in line 6, for example, “Torbern Tagesson g and h”.

2. There is a wrong punctuation in line 154.

3. There exists syntax error in line 230.

4. In references, the doi labelings in lines 412,476,478 were inconsistent with others.

Reviewer 3 Report

Ÿ   My main concern is that the conclusion should be more comprehensive and objective. From the experimental results, the authors compared the performance of the algorithm for a total of six months from April to October, and not all comparisons showed that MS was better than RGB, some months RGB was even better than MS, and the lead of MS over RGB was not very obvious (except for June 17), so how can this be judged that MS is better than RGB?

Ÿ   For data-driven deep learning algorithms, training data is crucial. However, I didn’t learn any detail information of the datasets. For example, how many figures are used? This is important for the my judgment of the results.

Ÿ   More training details of the DeepLabv3 in this study should be given.

Ÿ   Line 217, from Tab.4, it can be found that the RGB-based ones achieves a higher score compared with that of MS on April 21. Also on October 27. It needs to be explained or mentioned in the content.

Ÿ   Line 154-156, Large or Small is relative. Recommend changing this sentence to a more objective statement.

Ÿ   Line 193  what’s the P and R ?

Ÿ   Line 195, Where,……
